# Impact of Smoking Status in Combination Treatment with EGFR Tyrosine Kinase Inhibitors and Anti-Angiogenic Agents in Advanced Non-Small Cell Lung Cancer Harboring Susceptible EGFR Mutations: Systematic Review and Meta-Analysis

**DOI:** 10.3390/jcm11123366

**Published:** 2022-06-12

**Authors:** Tai-Huang Lee, Hsiao-Ling Chen, Hsiu-Mei Chang, Chiou-Mei Wu, Kuan-Li Wu, Chia-Yu Kuo, Po-Ju Wei, Chin-Ling Chen, Hui-Lin Liu, Jen-Yu Hung, Chih-Jen Yang, Inn-Wen Chong

**Affiliations:** 1Division of Pulmonary and Critical Care Medicine, Department of Internal Medicine, Kaohsiung Medical University Hospital, Kaohsiung Medical University, Kaohsiung 80708, Taiwan; weatlee@gmail.com (T.-H.L.); 1070476@kmuh.org.tw (K.-L.W.); 960225kmuh@gmail.com (P.-J.W.); jyhung@kmu.edu.tw (J.-Y.H.); 2Department of Internal Medicine, Kaohsiung Municipal Ta-Tung Hospital, Kaohsiung Medical University, Kaohsiung 80145, Taiwan; 3Department of Pharmacy, Kaohsiung Municipal Ta-Tung Hospital, Kaohsiung Medical University, Kaohsiung 80145, Taiwan; hlchen369@gmail.com (H.-L.C.); 880504@kmhk.org.tw (H.-M.C.); pompelmous23@gmail.com (C.-M.W.); 4Department of Internal Medicine, Kaohsiung Municipal Siaogang Hospital, Kaohsiung Medical University, Kaohsiung 81267, Taiwan; goba2356@gmail.com; 5Cancer Center, Kaohsiung Medical University Hospital, Kaohsiung Medical University, Kaohsiung 80708, Taiwan; 960645@kmuh.org.tw; 6Cancer Center, Kaohsiung Municipal Ta-Tung Hospital, Kaohsiung Medical University, Kaohsiung 80708, Taiwan; 0980610liu@gmail.com; 7School of Post-Baccalaureate Medicine, College of Medicine, Kaohsiung Medical University, Kaohsiung 80145, Taiwan

**Keywords:** epidermal growth factor receptor, EGFR, tyrosine kinase inhibitors, TKIs, bevacizumab, ramucirumab, apatinib, gefitinib, erlotinib, osimertinib

## Abstract

Patients with advanced non-small cell lung cancer (NSCLC) who harbor susceptible epidermal growth factor receptor (EGFR) mutations and are treated with EGFR tyrosine kinase inhibitors (TKIs) show longer progression-free survival (PFS) than those treated with chemotherapy. However, developed EGFR-TKI resistance limits PFS improvements. Currently, combination treatment with EGFR-TKIs and anti-angiogenic agents is considered a beneficial regimen for advanced-stage NSCLC harboring susceptible EGFR mutations. However, several trials reported osimertinib plus bevacizumab failed to show superior efficacy over osimertinib alone. However, subgroup analysis showed significantly longer PFS among patients with a history of smoking over those who never smoked. We performed a comprehensive systematic review and meta-analysis to evaluate the smoking status impact. At the end of the process, a total of 2068 patients from 11 randomized controlled trials (RCTs) were included in our meta-analysis. Overall, combination EGFR-TKI plus anti-angiogenic agent treatment showed significantly better PFS among patients with a smoking history (Hazard Ratio (HR) = 0.59, 95% confidence interval (CI) = 0.48–0.73). Erlotinib-based combination therapy showed positive PFS benefits regardless of smoking status (HR = 0.54, 95%CI = 0.41–0.71 for ever smoker, HR = 0.69, 95%CI = 0.54–0.87 for never smoker). Combination therapy prolonged PFS significantly regardless of ethnicity (HR: 0.64, 95% CI: 0.44–0.93 for Asian RCTs, HR: 0.55, 95% CI: 0.41–0.74 for global and non-Asian RCTs). PROSPERO registration number is CRD42022304198).

## 1. Introduction

Lung cancer remains the leading cause of cancer deaths worldwide, accounting for 1.8 million deaths each year globally [1]. Lung cancer can be categorized as either small cell lung cancer (SCLC) or non-SCLC (NSCLC). NSCLC accounts for 85% of all lung cancer cases and includes adenocarcinoma, squamous cell carcinoma, and large cell carcinoma [2]. More than 50% of patients with lung cancer are diagnosed at the advanced-stage, and the 5-year survival rate for metastatic NSCLC is below 10% [3]. No cure has been identified for late-stage NSCLC patients, making lung cancer the most devastating cancer type worldwide. In past decades, chemotherapy has been the major treatment option for advanced-stage NSCLC patients, but treatment efficacy has been limited.

Epidermal growth factor receptor (EGFR) is one of the members of ErbB/HER transmembrane receptor family, and the EGFR mutation facilitating cellular regulation, proliferation, apoptosis, and angiogenesis. The incidence of EGFR mutation in NSCLC is around 50% in Asian, and it was around 10–15% in western population [4,5,6]. The EGFR mutation was always detected only in adenocarcinoma, and had higher prevalence in women, never smoker, and Asian [6]. The development of EGFR tyrosine kinase inhibitors (TKIs) in the 2000s resulted in dramatic improvements in both the response rate (RR) and progression-free survival (PFS) over traditional chemotherapy for late-stage NSCLC patients who harbor susceptible EGFR mutations [7,8,9,10,11,12], and EGFR-TKIs have become the standard of care for these patients. For NSCLC patients harboring EGFR mutation, TKIs have shown favorable benefits for RR and PFS (10–12 months).

A history of smoking is associated with poor prognosis in advanced lung adenocarcinoma treated with chemotherapy [13]. Several possible mechanisms have been proposed to explain this difference; for example, smoking might decrease TKI serum levels or induce EGFR-TKI resistance through EGFR hyperactivation, c-MET overexpression, mediating Src activation, induction of the epithelial to mesenchymal transition (EMT), or increased ATP-binding cassette transporter G2 (ABCG2)-dependent drug efflux [14,15,16,17]. In addition, owing to the many carcinogens in tobacco smoke, smoking-related malignancies have a high genome-wide burden of mutations, including p53 and p53 protein, is the most frequently mutated tumor suppressor in cancer, and p53 mutations can be associated with primary or acquired resistance to EGFR-TKIs. A retrospective study of several solid malignancies indicated that the longer PFS on standard systemic therapy was significantly longer with bevacizumab-containing regimens in patients with mutant p53 tumors [18,19,20].

Zhang et al. showed non-smoking was associated with significant prolonged PFS (HR, 0.73, 0.60 to 0.88; *p* = 0.001) compared to ever smoking, but no obvious difference in objective response rate and disease control rate [21]. In NSCLC patients with EGFR mutation, a meta-analysis also showed never smokers had greater benefit while received first line EGFR TKI compared with smokers [22].

Although TKIs have shown favorable benefits for RR and PFS, acquired resistance typically develops in patients after 10–12 months of treatment [7,8,9,10,11,12]. To prolong PFS in cases with mutated NSCLC, synergistic combinations are often added to the initial EGFR-TKI treatment. Many studies have been designed to identify methods able to combat or delay the development of drug resistance, and several trials testing combinations between EGFR-TKIs and chemotherapy or anti-angiogenic agents have been proposed. 

Bevacizumab, the best-characterized anti-angiogenic agent, is a recombinant, humanized vascular endothelial growth factor (VEGF) monoclonal antibody. Bevacizumab exerts multiple effects, such as inducing the regression of existing tumor vasculature, inhibiting new vessel growth, and reducing the permeability of the surviving vasculature, contributing to improved treatment efficacy [23]. In the ECOG4599 trial, the combination of bevacizumab and platinum doublet chemotherapy demonstrated significant improvements in both PFS and overall survival (OS) in patients with adenocarcinoma compared with traditional doublet chemotherapy alone [24]. Furthermore, several preclinical studies demonstrated the efficacy of combination treatment with EGFR-TKIs and bevacizumab [25,26,27]. For example, in an *EGFR*-mutated NSCLC xenograft model, bevacizumab was shown to counteract the development of VEGF-dependent resistance against erlotinib [27]. The mechanisms responsible for the observed effects of combination bevacizumab and erlotinib were proposed to be the inhibition of tumor angiogenesis, the improved delivery of erlotinib to the tumor site through vascular normalization, the alleviation of immunosuppression, and the promotion of efficient tumor infiltration by effector immune cells. 

Several phase 2 and phase 3 clinical trials examining treatment with erlotinib combined with bevacizumab also demonstrated significantly longer PFS than EGFR-TKI treatment alone. Ramucirumab, a fully human IgG1 monoclonal antibody against VEGF receptors (VEGFRs), also demonstrated significantly longer PFS than erlotinib alone in the RELAY trial [28]. 

Recently, two trials presented at the European Society of Medical Oncology (ESMO) in 2021 failed to show clinical benefits for the combination of osimertinib and bevacizumab in NSCLC patients who harbor *EGFR* mutation when used as either a first-line (WJOG 9717L) [29] or second-line (ETOP 10-16 BOOSTER) [30] regimen, but both trials showed a significant improvement in PFS when combination therapy was used in the ever smoker subgroup. In addition, the BEVERLEY trial, which was presented at the ESMO in 2021, was also designed to compare erlotinib plus bevacizumab combination treatment with erlotinib alone in treatment-naïve patients and showed significant improvements among the ever smoker subgroup but no effects for never smokers [31].

The purpose of this study was to perform a systematic review and meta-analysis of currently available clinical trials to evaluate the efficacy of combination treatment using EGFR-TKIs and anti-angiogenic agents and to analyze the impact of smoking status among advanced-stage NSCLC patients who harbor EGFR mutations when treated with EGFR-TKIs alone or in combination with anti-angiogenic agents

## 2. Materials and Methods

This study was organized in accordance with the preferred reporting items for systematic reviews and meta-analyses (PRISMA). A prospective protocol was created in advance and registered on the PROSPERO website (registration number: CRD42022304198).

### 2.1. Search Strategy and Study Selection 

A comprehensive literature search was performed in PubMed, Embase, ClinicalTrials.gov (accessed on 19 May 2022) database and ICTRP registry up to the 30 November 2021 without language limitation. In addition, we also searched the conference abstracts from annual meeting of American Society of Clinical Oncology (ASCO), European Medical Oncology (ESMO), American association of cancer research (AACR) and the World Conference on Lung Cancer (WCLC). Medical Subject Heading (MeSH) terms or Emtree terms of “non-small cell lung cancer”, “tyrosine kinase inhibitor”, and “anti-angiogenic inhibitors” were applied as search keywords and detailed search strategy was presented in Appendix A. Inclusion criteria for this study were as follows: (1) completed phase II–IV clinical trial; (2) adult patient with EGFR-mutant advanced NSCLC; (3) active control in RCTs was EGFR-TKI monotherapy, such as gefitinib, erlotinib, icotinib, afatinib, and osimertinib; (4) combination therapy with EGFR-TKIs plus anti-angiogenic inhibitors; and (5) efficacy comparisons between combination therapy and monotherapy were published. 

### 2.2. Data Extraction and Quality Assessment

Data extraction and quality assessment were conducted by trained reviewers (HL Chen and TH Lee); in addition, discrepancies between two independent reviewers were resolved by the discussion with a third reviewer (CJ Yang). Study characteristics of were presented in Table 1, including name of RCT, name of author, study design, baseline character (age, gender, metastasis, stage of cancer, ECOG score, previous therapy, smoking status), intervention (medication, dosage, duration), and treatment outcomes. Quality judgments was assessed by the ‘Risk of bias’ assessment tool (ROB), which was taken from the Cochrane Handbook for Systematic Reviews of Interventions, Version 5.4. Seven domains were assessed in ROB tool, such random sequence generation, allocation concealment, blinding of participants and personnel, blinding of outcome assessment, incomplete outcome data, selective reporting, and other biases. 

### 2.3. Data Synthesis and Statistical Analysis

Overall survival (OS), progression free survival (PFS) and objective response rate (ORR) was evaluated to present the treatment efficacy. Due to OS and PFS was time dependent indicator, adjusted hazard ratio (HR) was regarded as effect size. In addition, response ratio was used as effect size for binary indicators, such as ORR. Meta analysis was conducted under the DerSimonian and Laird random effects model, which assume a common intervention effect is relaxed, and the effect sizes have a distribution. Heterogeneity was evaluated by chi-squared test and I-squared test. Chi-squared test assesses whether observed differences in results were compatible with chance alone and I-squared test interpreted the proportion of total variation due to heterogeneity rather than sampling error. Finally, we reported the publication bias by funnel plot, which was a simple scatter plot of the intervention effect estimates from individual studies with the size or precision. This meta-analysis was performed using Review Manager (RevMan) Version 5.4. which was developed by the Cochrane Collaboration.

## 3. Results

### 3.1. Literature Search

The search process used for the systematic reviews is presented in Figure 1 and was performed according to the 2020 Preferred Reporting Items for Systematic Reviews and Meta-Analyses (PRISMA) guidelines. A total of 366 studies or randomized control trials (RCTs) were imported from the PubMed, Embase, and RCT registry following the performance of automated searches. Of these studies, 111 duplications were removed, and 112 studies were excluded because their design, population, or intervention did not meet our established inclusion criteria. After title and abstract screening, 39 studies were included for full-text review. We updated data of each RCT; therefore, 15 midterm RCT reports were excluded because the updated results were published. In addition, 7 studies were excluded because anti-angiogenic agents were administered following EGFR-TKI therapy. Finally, 7 meta-analyses were removed after we reviewed the reference list for more relevant studies. After the full-text review, 10 studies were included in our analysis. We also identified 25 studies from the websites of global conferences and citations from the included RCTs. Among these, 12 duplications, 4 observed reports, and 3 protocols without results were excluded, resulting in 6 studies included. In total, 4 of them were duplicated with published studies from database, and the remaining 2 studies (BEVERLY trial and ETOP BOOSTER trial) were just presented in the conference abstract. At the end of the process, 16 published studies describing the results of 11 RCTs with 2068 patients were included in the in our meta-analysis.

### 3.2. Study Characteristics and Quality Evaluation

A total of 2068 patients from 11RCTs were included in our meta-analysis. Details regarding the characteristics of the 11 included RCTs are provided in Table 1. All included RCTs were conducted for patients with advanced or metastatic NSCLC harboring EGFR mutations. Most RCTs enrolled treatment-naïve patients, but the ETOP BOOSTER and WJOG 8715L trials also enrolled patients with acquired T790M mutations after initial EGFR-TKI treatment [30,36]. An osimertinib-based regimen was applied in 3 RCTs (ETOP BOOSTER, WJOG 8715L, and WJOG 9717L) [29,30,36], a gefitinib-based regimen was applied in 2 RCTs (BAGEL [37] and CTONG1706 [38]), and an erlotinib-based regimen was applied in the remaining trials.

The majority of the enrolled trials included in the meta-analysis used bevacizumab as the anti-angiogenic agent, but ramucirumab was used in the RELAY trial [24], and apatinib was used in the CTONG1706 trial [38]. The RELAY and ETOP BOOSTER trials recruited NSCLC patients from a global setting [28,30], The BEVERLY trial and Stinchcombe et al. recruited only Caucasian participants, and the remaining RCTs recruited Asian participants [31,35]. The median age was 64–74 years, and the percentage of men across the trials ranged from 17% to 40%. Patients with any smoking history, including former smokers, light smokers, and current smokers.

The results of the quality assessment are presented in Figure 2. The protocols for the BEVERLY, ETOP BOOSTER, and WJOG 9717L trials were not provided in the published reports [29,30,31]; therefore, we retrieved detailed information from the RCT registration website and the ESMO presentation data. Performance bias was assessed in most RCTs due to the open-label design. However, performance bias appeared to have limited influence because objective indicators, such as OS, PFS, and overall RR (ORR), were applied as our study outcomes.

### 3.3. Efficacy Comparisons between Combination Treatment with an EGFR-TKI plus an Anti-Angiogenic Agent and Treatment with an EGFR-TKI Alone

The results of the comparisons between combination EGFR-TKI plus anti-angiogenic agent therapy and EGFR-TKI monotherapy for OS, PFS, and ORR are shown in Appendix B. Compared with EGFR-TKI monotherapy, combination therapy with an EGFR-TKI and an anti-angiogenic agent significantly increased PFS (HR: 0.73, 95% CI: 0.60–0.88; Figure A1) but had no effect on OS (HR: 0.92, 95% CI: 0.79–1.07; Figure A2) or ORR (HR: 1.04, 95% CI: 0.99–1.09; Figure A3).

Among the different EGFR-TKIs tested, patients who received erlotinib plus an anti-angiogenic agent had significant benefits for PFS (HR: 0.60, 95% CI: 0.52–0.69) compared with erlotinib alone. However, no significant benefit was observed for OS (HR: 0.89, 95% CI: 0.76–1.05) or ORR (RR: 1.05, 95% CI: 0.98–1.13). Among patients treated with gefitinib, a superior effect was noticed for PFS in response to combination therapy compared with TKI monotherapy (HR: 0.71, 95% CI: 0.54–0.93) but not for OS (HR: 1.10, 95% CI: 0.72–1.68) or ORR (RR: 1.05, 95% CI: 0.92–1.19). No benefits were observed for PFS (HR: 1.08, 95% CI: 0.80–1.47), OS (HR: 0.97, 95% CI: 0.51–1.86), or ORR (RR: 1.01, 95% CI: 0.88–1.15) for osimertinib combined with an anti-angiogenic agent compared with osimertinib alone.

### 3.4. Treatment Effects among Ever Smokers and Never Smokers

The study results were then stratified according to smoking status. Former smokers, light smokers, and current smokers were categorized as ever smokers. Subjects with no history of tobacco use were categorized as never smokers.

Figure 3 shows that combination treatment with an EGFR-TKI plus an anti-angiogenic agent showed significantly longer PFS than treatment with an EGFR-TKI alone (HR: 0.74, 95% CI: 0.61–0.89), and a larger increase in PFS was observed for ever-smokers (HR: 0.59, 95% CI: 0.48–0.73) using combination therapy compared with EGFR-TKI monotherapy. However, the combination regimen did not show superior efficacy compared with EGFR-TKI alone in never smokers (HR: 0.89, 95% CI: 0.67–1.19). Finally, no asymmetry was presented in the funnel plots (Appendix C), so the publication bias was limited.

### 3.5. Treatment Effects among Ever Smokers and Never Smokers Stratified by Treatment Regimen

The combined analysis of all trials for the ever smoker group is shown in Figure 4. Combination therapy using an EGFR-TKI plus am anti-angiogenic agent showed significantly improved PFS compared with any EGFR-TKI alone (HR: 0.59, 95% CI: 0.48–0.73). Among the various EGFR-TKI–based regimens, erlotinib- and osimertinib-based regimens significantly improved PFS in ever smokers, whereas the gefitinib-based regimens failed to demonstrate superior efficacy, regardless of smoking status.

Figure 5 shows the overall effect of combination therapy using an EGFR-TKI plus an anti-angiogenic agent among never smokers, which failed to show an improved PFS compared with the use of am EGFR-TKI alone (HR: 0.89, 95% CI: 0.67–1.19). Analysis of individual EGFR-TKI–based regimens revealed that erlotinib-based combination regimens showed significantly longer PFS (HR: 0.69, 95% CI: 0.54–0.87), but gefitinib- and osimertinib-based combination regimens failed to show superior efficacy compared with either EGFR-TKI alone. The combination of osimertinib plus an anti-angiogenic agent failed to show a longer PFS for the entire enrolled population (Appendix B, HR: 1.08, 95% CI: 0.80–1.47) and for never smokers (HR: 1.41, 95% CI: 1.00–1.99), but osimertinib-based combination regimens presented significantly longer PFS among ever smokers (HR: 0.64, 95% CI: 0.42–0.99) in the meta-analysis.

### 3.6. Treatment Effects among Ever Smokers and Never Smokers Stratified by Ethnicity

Among ever smoker, combination therapy prolonged PFS significantly regardless of ethnicity in Figure 6 (HR: 0.59, 95% CI: 0.48–0.73). Asian RCTs (HR: 0.64, 95% CI: 0.44–0.93) demonstrated less PFS benefit than global and non-Asian RCTs (HR: 0.55, 95% CI: 0.41–0.74). In terms of never smoker (Figure 7), no superior effects were presented for patients received combination therapy among Asian RCTs (HR: 0.92, 95% CI: 0.56–1.51) and global and non-Asian RCTs (HR: 0.90, 95% CI: 0.61–1.34).

## 4. Discussion

In the meta-analysis, we discovered that combination treatment with an EGFR-TKI plus an anti-angiogenic agent significantly increased PFS with no observed effects on OS or ORR compared with EGFR-TKI treatment alone. Patients treated with erlotinib or gefitinib combined with an anti-angiogenic agent showed significant benefits for PFS but no significant benefits in OS or ORR compared with either EGFR-TKI alone. In addition, the combination of an anti-angiogenic agent with osimertinib had no effect on ORR, PFS, or OS relative to osimertinib alone.

In our meta-analysis, we firstly aim to analyze the impact of smoking status on all enrolled RCTs consisting of an EGFR-TKI and an anti-angiogenic agent. In addition to cost, antiangiogenic agents have some adverse drug reactions, such as hypertension, proteinuria, epistaxis, gastrointestinal bleeding, and so on. Therefore, we have to identify who are the most beneficial population receiving the combination therapy of EGFR TKI and antiangiogenic agent.

The combination of an EGFR-TKI and an anti-angiogenic agent produced beneficial effects for PFS compared with EGFR-TKI treatment alone among ever smokers but not among never smokers. However, osimertinib-based combination regimens failed to show superior efficacy compared with osimertinib alone. Erlotinib-based combination therapies significantly improved PFS regardless of smoking status, whereas gefitinib-based combination therapy showed no effects in both never smokers and ever smokers.

In the targeted therapy era, first-generation EGFR-TKI treatment resulted in longer PFS than traditional chemotherapy in NSCLC patients harboring susceptible EGFR mutations. Acquired resistance to first- or second-generation EGFR-TKI occurs in almost all patients. Therefore, improving PFS among patients with acquired resistance represents an urgent need, leading to the development of EGFR-TKI–based combination treatments.

Although several clinical trials have shown positive results, the mechanisms through which bevacizumab and erlotinib combination therapy results in improved efficacy remain unknown. The VEGF pathway has been recognized as a key mediator of angiogenesis, which is necessary to support tumorigenesis, and EGFR activation has been shown to upregulate VEGF levels [39]. When the EGFR pathway is inhibited, VEGF may be downregulated. In one study, an EGFR-mutated NSCLC xenograft model was treated with erlotinib until resistance was detected, which revealed decreased VEGF levels after initial erlotinib treatment and increased VEGF levels after the development of erlotinib resistance [27]. Preclinical studies reported that the combination of an EGFR-TKI plus a VEGF inhibitor resulted in a synergistic effect able to partially reverse resistance to EGFR-TKI treatment in xenograft models harboring EGFR mutations [25,26,27].

Initially, the phase 3 BeTa lung study was designed to compare erlotinib combined with bevacizumab against erlotinib alone as a second-line treatment for patients with advanced NSCLC and showed that combination treatment had no benefits in terms of PFS (HR: 0.97, 95% CI: 0.80–1.18, *p* = 0.7583) [40]. However, a significantly longer PFS was observed in NSCLC patients with mutated EGFR in response to combination treatment.

Furthermore, the phase 2 JO25567 trial conducted in Japan enrolled more treatment-naïve patients with stage IIIB and IV NSCLC harboring activating EGFR mutations and showed a significantly longer PFS among patients who received combination erlotinib and bevacizumab than among those who received erlotinib alone [33]. Based on these results, the phase 3 NEJ026 trial was conducted in Japan using the same study design, which also showed a significant improvement in PFS among patients treated with bevacizumab plus erlotinib combination therapy than among patients treated with erlotinib alone (16.9 months vs. 13.3 months, HR: 0.605, 95% CI: 0.417–0.877; *p* = 0.016) [34]. The ARTEMIS (CTONG 1509) trial, conducted in China, used a similar design as NEJ026 and showed a significant improvement in PFS (18 months vs. 11.3 months, HR: 0.55 *p* < 0.001) [32]. The phase 3 RELAY trial was designed to compare ramucirumab plus erlotinib and erlotinib alone and ramucirumab plus erlotinib showed superior PFS in patients with EGFR-mutated metastatic NSCLC (19.4 months vs. 12.4 months; HR: 0.59; *p* < 0.0001) [28].

Apatinib, a potent VEGFR2-TKI, specifically binds to VEGFR2 and induces anti-angiogenetic and anti-neoplastic effects. The ACTIVE (CTONG1706) trial in China was designed to compare apatinib plus gefitinib against gefitinib alone [38]. The PFS for apatinib plus gefitinib improved compared with the PFS for gefitinib alone (13.1 months vs. 10.2 months, HR: 0.71, 95% CI: 0.54–0.93, *p* = 0.02). In addition, gefitinib combined with bevacizumab showed reduced PFS compared with gefitinib alone in the BAGEL study by Kitagawa et al. [37]. We showed a longer PFS for gefitinib-based combination regimens, but the effect was not prominent when we divided enrolled patients into ever smokers and never smokers, which may be due to the small sample size.

The selection of patients who should receive anti-angiogenic agents in addition to standard EGFR-TKI therapy for metastatic NSCLC harboring EGFR mutations is an important issue, requiring several points of consideration. First, the advantages and disadvantages associated with the use of anti-angiogenic agents must be addressed. Although anti-angiogenic agents combined with EGFR-TKIs demonstrated beneficial effects for PFS in our meta-analysis, anti-angiogenic agents may also cause adverse reactions, such as epistaxis, hypertension, proteinuria, bleeding, and tracheoesophageal fistula [41]. Smoking may hamper the efficacy of EGFR-TKIs in patients with EGFR-mutated NSCLC, and the overall effect of combination EGFR-TKI and anti-angiogenic agents in ever smokers showed significant improvement in the WJOG9717L, ETOP 10-16 BOOSTER, and BEVERLEY trials [29,30,31]. Our meta-analysis indicated that ever smokers received positive benefits when treated with osimertinib plus bevacizumab, and erlotinib plus bevacizumab showed beneficial effects regardless of smoking status. The most appropriate treatment for various populations of patients with EGFR-mutated NSCLC should be based on the availability of EGFR-TKIs, anti-angiogenic agents, and smoking status. Our comprehensive meta-analysis demonstrated that smoking status might play a key role in decision-making when selecting combinations of different EGFR-TKIs with anti-angiogenic agents or EGFR-TKIs alone.

In conclusion, our study indicated that the combination therapy consisting of an EGFR-TKI with an anti-angiogenic agent demonstrated significantly longer PFS and OS for erlotinib-based regimens but not for osimertinib-based regimens. The combination regimen of osimertinib plus bevacizumab showed significant improvement in ever smokers but not in never smokers. We believe our meta-analysis study, which focuses on the differential effects observed when different EGFR-TKIs are combined with anti-angiogenic agents in the context of smoking status, provides very useful information that can be applied in clinical settings to improve patient outcomes.

This meta-analysis has many limitations. First, only 11 RCT trials were enrolled in this meta-analysis, indicating the need for additional, well-designed RCTs to validate these conclusions. Second, large variations in smoking habits existed among the ever smoker cohort in this study, and different smoking habits may result in different clinical impacts. Third, the smoking status of patients in the enrolled RCT was not well-documented. Smoking status was a subjective description, with some trials dividing smokers into never smokers, former smokers, light smokers, and current smokers, and some trials divided smoking status into ever and never smokers. In addition, accurate descriptions of smoking amounts were lacking. The unknown or poor definition of smoking status limits the generalizability of the meta-analysis results to real-world conditions. Fourth, ramucirumab-based treatment was only tested in the RELAY trial [28], and apatinib was only used in the CTONG1706 trial [38], whereas all remaining trials used bevacizumab as the anti-angiogenic agent; therefore, we cannot make definite conclusions regarding whether all anti-angiogenic agents combined with EGFR-TKIs provide similar benefits for PFS, although the overall effect of the meta-analysis suggested a beneficial effect for the combination regimens.

Future research exploring the combination EGFR-TKIs plus anti-angiogenic agents and EGFR-TKIs alone should carefully document more accurate smoking status among enrolled patients to improve the reliability of results which could improve the clinical application of these therapies in patients with EGFR-mutated NSCLC. Large-scale RCTs designed to examine the impacts of smoking status on the outcomes of combination or monotherapy remain necessary to validate our conclusions.

## 5. Conclusions

The combination of EGFR-TKIs and anti-angiogenic agents are popular regimens for the treatment of advanced-stage NSCLC harboring susceptible EGFR mutations. However, not all regimens result in beneficial effects in clinical practice. Our meta-analysis indicated that the overall effect of combination therapy consisting of EGFR-TKIs plus anti-angiogenic agents is beneficial for PFS, but not for OS or ORR. Erlotinib-based combination regimens demonstrated significantly longer PFS in both ever smokers and never smokers. However, osimertinib-based combination regimens were not superior to osimertinib alone for treatment-naïve patients or those with acquired T790M resistance mutations but did show improved PFS for smokers in this meta-analysis.

## Figures and Tables

**Figure 1 jcm-11-03366-f001:**
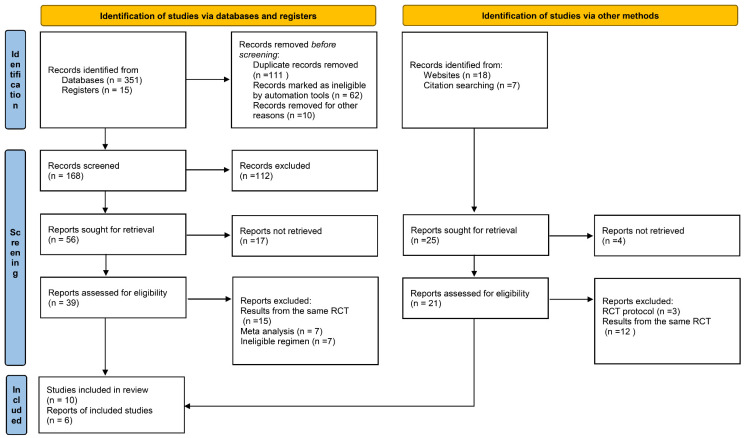
PRISMA 2020 flow diagram for new systematic reviews.

**Figure 2 jcm-11-03366-f002:**
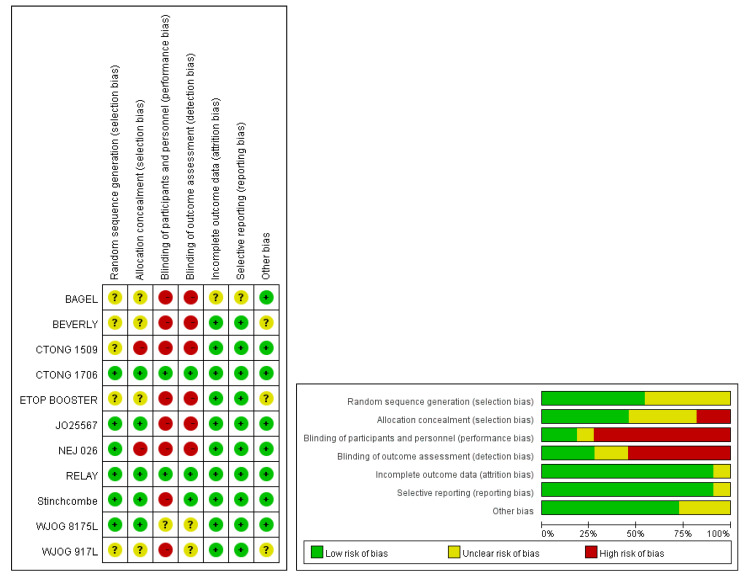
Risk of bias assessment.

**Figure 3 jcm-11-03366-f003:**
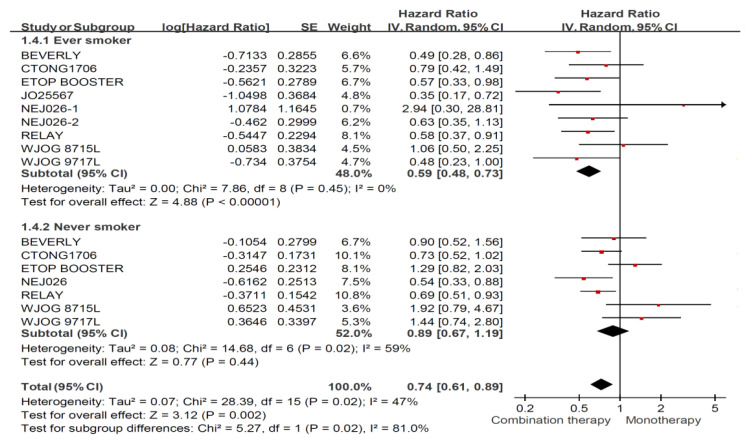
Forest plot and pooled hazard ratio for progression free survival (PFS) among all enrolled patients (combination therapy vs. monotherapy).

**Figure 4 jcm-11-03366-f004:**
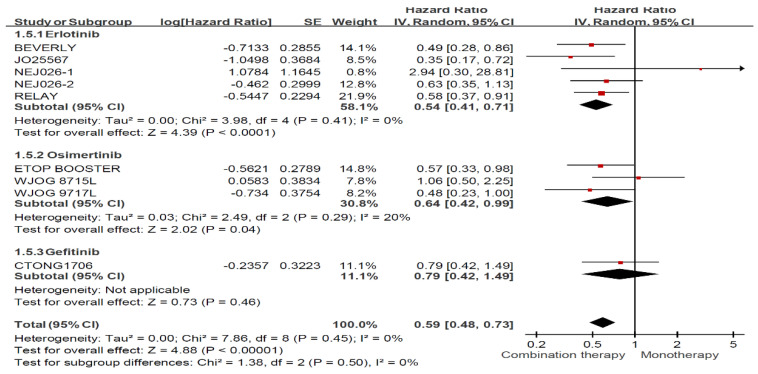
Forest plot and pooled hazard ratio for progression free survival (PFS) in the ever smoker subgroup (Stratified by regimen).

**Figure 5 jcm-11-03366-f005:**
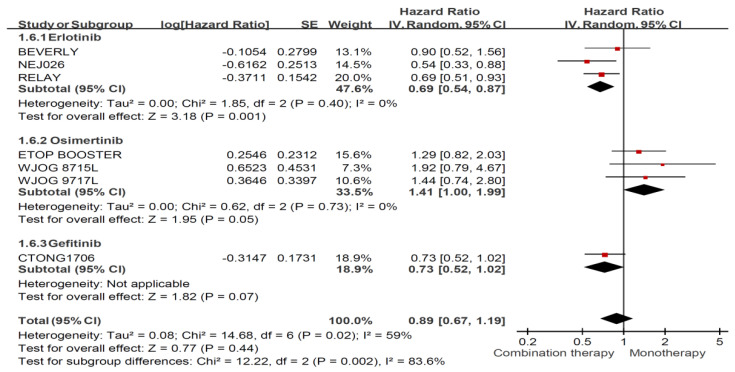
Forest plot and pooled hazard ratio for progression free survival (PFS) in the never smoker subgroup (Stratified by regimen).

**Figure 6 jcm-11-03366-f006:**
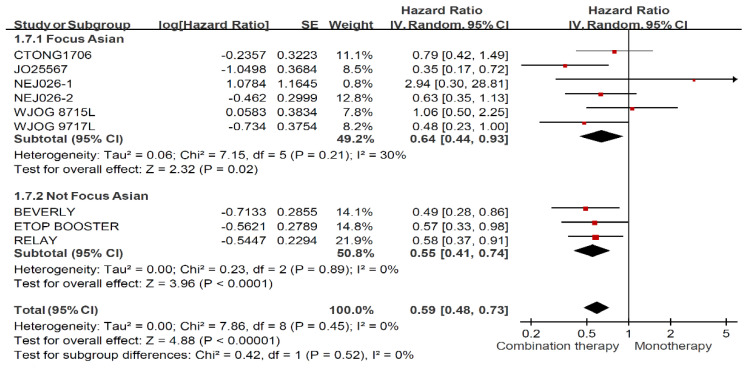
Forest plot and pooled hazard ratio for progression free survival (PFS) in the ever smoker subgroup (Stratified by ethnicity).

**Figure 7 jcm-11-03366-f007:**
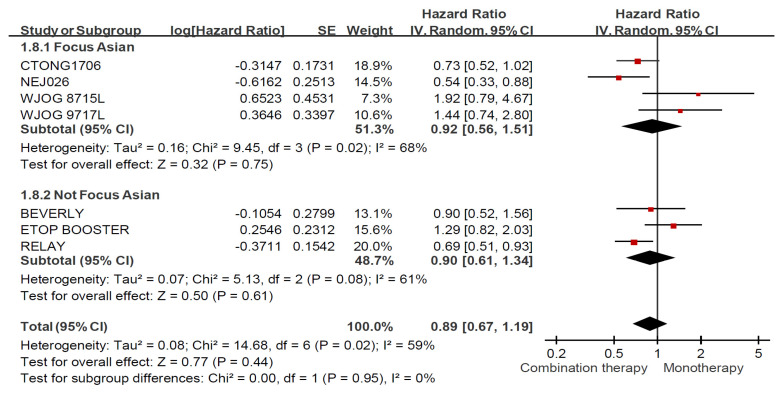
Forest plot and pooled hazard ratio for progression free survival (PFS) in the never smoker subgroup (Stratified by ethnicity).

**Table 1 jcm-11-03366-t001:** Characteristics of included studies.

Trial	BEVERLY [31]		CTONG 1509 [32]		JO25567 [33]		NEJ 026 [34]		RELAY [28]		Stinchcombe [35]		ETOP BOOSTER [30]		WJOG 8175L [36]		WJOG 917L [29]		KITAGAWA [37]		CTONG 1706 [38]	
**Author**	**Maria et al.**		**Zhou et al.**		**Yamamoto et al.**		**Saito et al.**		**Nakagawa et al.**		**Stinchcombe et al.**		**Soo et al.**		**Toi et al.**		**Kenmotsu et al.**		**Kitagawa et al.**		**Zhao et al.**	
**Year**	2021		2019		2021		2019 PFS;		2019		2019		2021 ESMO poster		2020 ESMO		2021 ESMO		2019		2021	
	ESMO						2020 OS															
**Design**	Phase 3 RCT		Phase 3 RCT		Phase 2 RCT		Phase 3 RCT		Phase 3 RCT		Phase 2 RCT		Phase 2 RCT		Phase 2 RCT		Phase 2 RCT		Phase 2 RCT		Phase 3 RCT	
**Intervention**	**Erlotinib**	**Erlotinib + Bevacizumab**	**Erlotinib**	**Erlotinib + Bevacizumab**	**Erlotinib**	**Erlotinib + Bevacizumab**	**Erlotinib**	**Erlotinib+ Bevacizumab**	**Erlotinib**	**Erlotinib + Ramuicirumab**	**Erlotinib**	**Erlotinib + Ramuicirumab**	**Osimertinib**	**Osimertinib + Ramuicirumab**	**Osimertinib**	**Osimertinib + bevacizumab**	**Osimertinib**	**Osimertinib + bevacizumab**	**Gefitinib**	**Gefitinib + Bevacizumab**	**Apatinib + Gefitinib**	**Placebo + Gefitinib**
**Sample size**	80	80	154	157	77	75	112	112	221	225	45	43	77	78	41	40	61	61	10	6	157	156
**Patient character**																						
**Age (median)**	67.7	65.9	57	59	67	67	68	67	64	65	63	65	67		70	68	66	67	72.5	73.5	57	60
**Male (%)**	38%	35%	38%	38%	34%	40%	35%	37%	37%	37%	31%	28%	38%		41%	40%	38%	39%	30%	17%	42%	40%
**ECOG 0~1 (%)**	95%	98%	100%	100%	100%	100%	99%	100%	100%	100%	100%	100%	NA		100%	100%	100%	100%	100%	%	100%	100%
**Brain metastasis (%)**	NA	NA	31%	28%	NA	NA	32%	32%	NA	NA	31%	26%	NA		22%	30%	NA	NA	NA	NA	33%	26%
**Stage IV (%)**	94%	96%	86%	90%	81%	80%	75%	73%	84%	87%	100%	100%	98%		63%	83%	75%	79%	90%	100%	97%	95%
**Smoking, ever (%)**	54%	43%	NA	NA	42%	43%	43%	42%	32%	29%	40%	49%	40%		49%	48%	51%	38%	20%	33%	27%	22%
**Outcome**	8																					
**PFS (months, median)**	9.7	15.4	11.3	18	9.8	16.4	13.3	16.9	12.4	19.4	13.5	17.9	12.3	15.4	13.5	9.4	20.2	22.1	15.1	5.4	13.7	10%
**PFS (HR, 95% CI )**	0.60 (0.42–0.85)		0.55 (0.41–0.75)		0.52 (0.35–0.76)		0.61 (0.41, 0.88)		0.59 (0.46–0.76)		0.81 (0.50–1.31)		0.96 (0.69–1.36)		1.44 (1.00–2.07)		0.86 (0.53–1.40)		NA		0.71 (0.54–0.95)	
**OS (months, median)**	23	28.4	NA	NA	47	47.4	46.2	50.7	NA	NA	50.6	32.4	24.3	24	22.1	NR	NA	NA	NA	NA	Not mature	Not mature
**OS (HR, 95% CI )**	0.70, (0.46–1.10)		NA		0.81 (0.53–1.24)		1.00 (0.68–1.47)		NA		1.41 (0.71–2.80)		HR 1.03; (95% CI 0.67–1.56; *p* = 0.91)		*p* = 0.96		NA		NA		HR for OS was 1.10 (95% CI: 0.72–1.67, *p* = 0.66).	
**ORR (%)**	50.00%	70.00%	84.70%	86.30%	64.00%	69.00%	66.00%	72.00%	76.00%	75.00%	83.00%	81.00%	55.00%	55.00%	5400.00%	6800.00%	86.00%	82.00%	44.00%	50.00%	77.10%	73.70%
**AE, Gr >= 3; (%)**	NA	NA	26%	55%	53%	91%	46%	88%	54%	72%	NA	NA	18%	47%	NA	NA	48%	56%	NA	NA	84%	38%
**Discontinued due to AE (%)**	NA	NA	3%	7%	18%	16%	7%	29%	11%	13%	NA	26%	4%	25%	31%	35%	26.70%	55.70%	NA	NA	29%	5%

Acronym: AE, Adverse events; ECOG, Eastern Cooperative Oncology Group; NA, not available; ORR, overall response rate; OS, overall survival; PFS, progression free survival; RCT, randomized controlled trial.

## Data Availability

Not applicable.

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
