# Peer review of "Impact of Smoking Status in Combination Treatment with EGFR Tyrosine Kinase Inhibitors and Anti-Angiogenic Agents in Advanced Non-Small Cell Lung Cancer Harboring Susceptible EGFR Mutations: Systematic Review and Meta-Analysis"

_jcm, 2022, doi:10.3390/jcm11123366_

Round 1
Reviewer 1 Report
Dear Authors,
I have read your article titled "Impact of Smoking Status in Combination Treatment with EGFR Tyrosine Kinase Inhibitors and Anti-Angiogenic Agents in Advanced Non–Small Cell Lung Cancer Harboring Susceptible EGFR Mutations: Systematic Review and Meta-Analysis" with interest and attention. The article design, methodology, discussion of the analyzes, tables and graphics were prepared very well.
Best Regards.
Author Response
Thanks for your kind words.
Reviewer 2 Report
The authors analyzed in their review and metanalysis the role of Impact of Smoking Status in Combination with treatments based on 2 EGFR Tyrosine Kinase Inhibitors and Anti-Angiogenic Agents. It is an interesting paper, but presents some points that may be improved.
Minor comments:
Figure 1 resolution very low and it is not legible.
Table 1 is not completed and included many grammatical mistakes
Table 1 resolution is very low.
Figure 2 resolution very low and it is not legible. Moreover it is not clear which studies were included
Regarding table 1, stage IV patients were not the 100% of include patients, which other stages were included in the different studies?
In discussion, it may be interesting to show the complications rate related to anti-angiogenic agent and the percentage in the considered studies.
In limitations the author reported 11 included studies, but in the results they were 10.
Author Response
Q1. Figure 1 resolution is very low and it is not legible.
Reply. Thanks for your kind suggestion. We have uploaded a revised Figure1 with a higher resolution.
Q2. Table 1 is not completed and included many grammatical mistakes
Reply. Thanks for your reminder. We have corrected the grammatical mistakes. According to figure1, 6 reports were identified via another resource (ex. website of global lung conferences). Among them, 5 reports were also published as full papers and identified via database. Therefore, 16 identified studies in Figure 1 (10 from the database, 6 from another resource) describe the results of 11 RCTs (presented in table1). This is the reason for the difference between figure1 and table1
Q3. Table 1 resolution is very low
Reply. We have uploaded a revised table1 with a higher resolution
Q4. Figure 2 resolution is very low and it is not legible. Moreover, it is not clear which studies were included?
Reply. We have uploaded a revised Figure2 with higher resolution, we included 11 RCTs in the revised figure2.
Q5 Regarding table 1, stage IV patients were not 100% of included patients, which other stages were included in the different studies?.
Reply. EGFR TKI was the golden standard treatment for advanced lung adenocarcinoma patients harbored with EGFR mutation. Therefore, most enrolled patients were stage IV but some were unresectable stage IIIB and stage IIIC.
Q6. In discussion, it may be interesting to show the complications rate related to the anti-angiogenic agents and the percentage in the considered studies.
Reply. As in previous studies, the TKI combined with an anti-angiogenic agent showed increased adverse events. The most common adverse effects were hypertension, skin rashes, and proteinuria. We summarized the studies result in a new Table
Q7. In limitations, the author reported 11 included studies, but in the results, they were 10.
Reply. Our study report included 11 studies; All studies were used for ORR calculation.